# Neural Implicit Manifold Learning for Topology-Aware Density Estimation

## Abstract

Natural data observed in $\mathbb{R}^n$ is often constrained to an $m$-dimensional manifold $\mathcal{M}$, where $m < n$. Current probabilistic models learn this manifold by mapping an $m$-dimensional latent variable through a neural network $f_\theta : \mathbb{R}^m \to \mathbb{R}^n$. Such procedures, which we call *pushforward models*, incur a straightforward limitation: manifolds cannot in general be represented with a single parameterization, meaning that attempts to do so will incur either computational instability or the inability to learn probability densities within the manifold. To remedy this problem, we propose to model $\mathcal{M}$ as a *neural implicit manifold*: the set of zeros of a neural network. To learn the data distribution within $\mathcal{M}$, we introduce *constrained energy-based models*, which use a constrained variant of Langevin dynamics to train and sample within a learned manifold. The resulting model can be manipulated with an *arithmetic of manifolds* which allows practitioners to take unions and intersections of model manifolds. In experiments on synthetic and natural data, we show that constrained EBMs can learn manifold-supported distributions with complex topologies more accurately than pushforward models.

## 1 Introduction

Here we focus on the common statistical task of estimating an unknown probability distribution $P^*$ using a set of datapoints $\{x_i\} \subset \mathbb{R}^n$ sampled from $P^*$. Commonly, the distribution of interest lies on an $m$-dimensional Riemannian submanifold $\mathcal{M}$ embedded in the ambient space $\mathbb{R}^n$, with $m < n$. For example, data from engineering or the natural sciences can be manifold-supported due to smooth physical constraints (Mardia et al., 2007; Boomsma et al., 2008; Brehmer & Cranmer, 2020). In general, the underlying submanifold $\mathcal{M}$ may be unknown *a priori*, which calls for us to design models which learn $\mathcal{M}$ in the process of learning $P^*$.

The typical paradigm for modelling distributions on learned manifolds is a *pushforward model*: a neural parameterization $f_\theta : \mathbb{R}^m \to \mathbb{R}^n$ trained to transform an $m$-dimensional prior into a flexible distribution on the data manifold embedded in $\mathbb{R}^n$ (e.g. Arjovsky et al. (2017); Tolstikhin et al. (2018); Arbel et al. (2021)). These techniques can generate high-resolution images, but are insufficiently flexible for learning distributions in settings where the true manifold structure is of interest.

Modelling a manifold as the image of a single mapping $f_\theta$ is topologically restrictive. For example, many approaches encourage an encoder $g_\phi$ and decoder $f_\theta$ to mutually invert each other at each datapoint (e.g. Donahue et al. (2017); Dumoulin et al. (2017); Xiao et al. (2019)), an objective we can precisely reinterpret as training $f_\theta$ to become a diffeomorphism between $\mathcal{M}$ and a subset of the latent space $\mathbb{R}^m$. This specification conflicts with the fact that, in general, $\mathcal{M}$ may have a complex topology which is not diffeomorphic to any such subset, exposing $f_\theta$ to a frontier of tradeoffs between expressivity and numerical stability (Cornish et al., 2020; Behrmann et al., 2021; Salmona et al., 2022). Even when $f_\theta$ is not a diffeomorphism, its continuity dictates many topological properties of the model manifold, such as connectivity and the number of holes (Munkres, 2000).

In this paper we learn data manifolds with a much broader class of topologies using a novel approach outlined in Figure 1. We first learn a manifold *implicitly* as the zero set of a neural network $F_\theta$, controlling the manifold dimension by regularizing the rank of its Jacobian. We then model the density within the manifold using a *constrained energy-based model* $E_\psi$, which uses constrained Langevin dynamics to sample points on the learned manifold. We show that constrained energy-based models on manifolds can be composed with each other akin to standard energy-based models (Hinton,

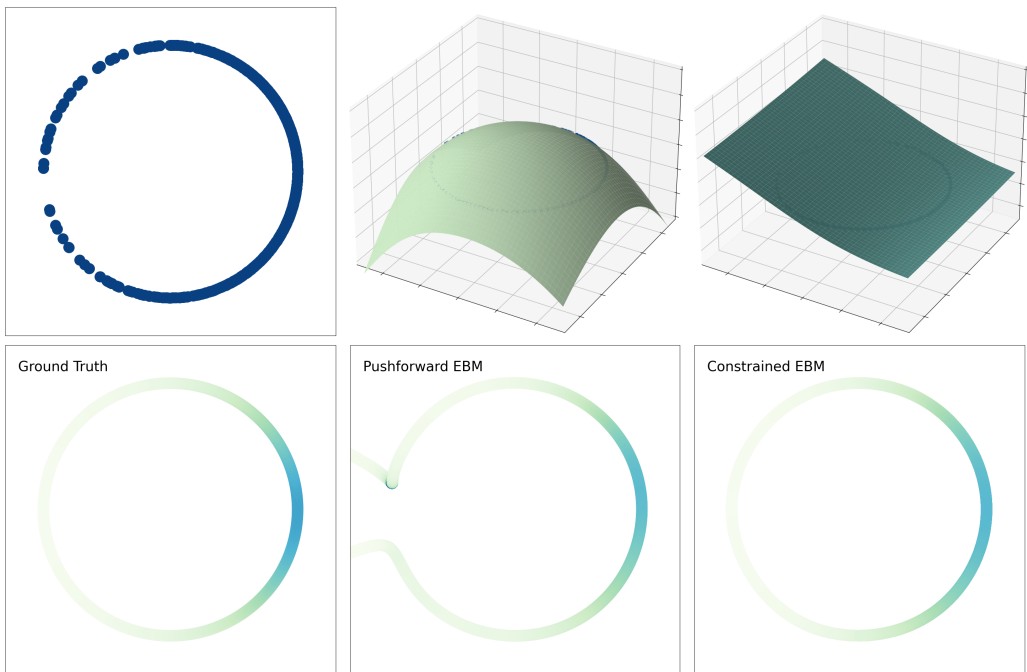

Figure 1: **In the top row**, our method is depicted on simulated circular data from a von Mises distribution. From left to right: ground truth sample of von Mises data, a manifold-defining function $F_\theta$ learned from the data, and an ambient energy $E_\psi$ trained with constrained Langevin dynamics on the learned manifold. **In the bottom row**, manifold learning and density estimation results from the resulting model are juxtaposed with a pushforward baseline. From left to right: the ground truth, a pushforward energy-based model, and a constrained energy-based model (ours). By defining the manifold with a constraint $F_\theta(x) = 0$, our method can model data with non-trivial topologies.

2002): manifold-defining functions $F_\theta$ along with their energies $E_\psi$ can be combined to take unions and intersections of data manifolds in what we call *manifold arithmetic*. We demonstrate theoretically and empirically that the proposed model can learn manifold-supported distributions more accurately than the pushforward paradigm prevalent in the current literature.

## 2 BACKGROUND AND RELATED WORK

### 2.1 MODELLING MANIFOLD-SUPPORTED DATA

**Manifold structure**    As above, suppose $\{x_i\}$ is a set of samples drawn from probability measure $P^*$ supported on $\mathcal{M}$, an $m$-dimensional Riemannian submanifold of $\mathbb{R}^n$. We focus on the case where $m < n$, so that $\mathcal{M}$ is "infinitely thin" in $\mathbb{R}^n$, meaning $P^*$ does not admit a probability density with respect to the standard Lebesgue measure. However, we may assume it has a density $p^*(x)$ with respect to the Riemannian measure of $\mathcal{M}$. We elaborate on this setup in Appendix A.

Models for manifold-supported data have long been of interest in statistics, machine learning, and various applications (Diaconis et al., 2013; McInnes et al., 2018). In particular, a number of past works have explored Monte Carlo methods on manifolds (Brubaker et al., 2012; Byrne & Girolami, 2013; Zappa et al., 2018), which we put to use here. However, the problem of simultaneously learning a submanifold *and* an underlying density has only become of interest in tandem with recent advances in deep generative modelling (Brehmer & Cranmer, 2020). To our knowledge, all such models fall under the umbrella of *pushforward models*.

**Density estimation with pushforward models**    When manifold-supported, $P^*$ is most commonly modelled as the *pushforward* of some latent distribution:

$$z \sim p_\psi(z), \quad x = f_\theta(z), \tag{1}$$

where $f_\theta : \mathbb{R}^m \to \mathbb{R}^n$ is a smooth mapping given by a neural network and $z \sim p_\psi(z)$ is a (possibly trainable) prior on $m$-dimensional latent space. The resulting model distribution $P_{\theta,\psi}$ is supported on the model manifold[1] $\mathcal{M}_\theta := f_\theta(\mathbb{R}^m)$. This framework encompasses generative adversarial networks (GANs) (Goodfellow et al., 2014; Arjovsky et al., 2017), injective flows (Brehmer & Cranmer, 2020; Caterini et al., 2021), and various regularized autoencoders (Makhzani et al., 2016; Tolstikhin et al., 2018; Ghosh et al., 2020; Kumar et al., 2020). Since we take the support to be an $m$-dimensional submanifold, we rule out bijective normalizing flows (Rezende & Mohamed, 2015; Dinh et al., 2017) and variational autoencoders (VAEs) (Kingma & Welling, 2014), unless $p_\theta(x|z)$ is a point mass.

In recent work, Loaiza-Ganem et al. (2022) outline a general procedure for manifold learning and density estimation with pushforward models, which separates modelling into two components: a *generalized autoencoder*, which embeds the data manifold into $m$-dimensional latent space, and a *density estimator*, which learns the density within the manifold. The generalized autoencoding step treats $f_\theta$ as a decoder, pairing it with a smooth encoder $g_\phi : \mathbb{R}^n \to \mathbb{R}^m$, and trains them to learn $\mathcal{M}$ by mutually inverting each other on the data,[2] such as by minimizing a reconstruction loss $\mathbb{E}_{x \sim P^*} ||x - f_\theta(g_\phi(x))||^2$. The density estimator $p_\psi(z)$ is then fitted to the encoded data $\{g_\phi(x_i)\}$ via maximum-likelihood. Given a datapoint $x \in \mathcal{M}$, two-step models estimate $p^*(x)$ as follows:

$$p_{\theta,\psi}(x) = p_\psi(z) \left| \det J_{f_\theta}^\top(z) J_{f_\theta}(z) \right|^{-1/2}, \tag{2}$$

where $z := g_\phi(x)$ is the encoding of $x$ and $J_{f_\theta}$ is the Jacobian of $f_\theta$ with respect to its inputs $z$. The fidelity of this estimate depends on the condition $f_\theta(g_\phi(x)) = x$ for all $x \in \mathcal{M}$; in other words, $g_\phi$ must be a right-inverse of $f_\theta$ on $\mathcal{M}$. Injective flow models (Brehmer & Cranmer, 2020; Caterini et al., 2021; Kothari et al., 2021; Ross & Cresswell, 2021) enforce invertibility on $\mathcal{M}_\theta$ with architectural constraints; other two-step models (Xiao et al., 2019; Ghosh et al., 2020; Rombach et al., 2022), like Loaiza-Ganem et al. (2022), achieve this condition at their non-parametric optimum.

**Topological challenges**  Despite the broad applicability of this density estimation procedure, the requisite right-invertibility condition is *effectively impossible* to satisfy for general manifolds $\mathcal{M}$. If $f_\theta(g_\phi(x)) = x$ for all $x \in \mathcal{M}$, then by definition, $g_\phi$ smoothly embeds $\mathcal{M}$ into $\mathbb{R}^m$. This condition presents an immediate topological challenge: $\mathcal{M}$ is an $m$-dimensional manifold, which in general cannot be embedded in $m$-dimensional Euclidean space. In line with the *strong Whitney embedding theorem* (Lee, 2013, pg.135), $\mathcal{M}$ might not be embeddable in Euclidean space of less than $2m$ dimensions.[3] It is thus impossible in the general case for the support of the prior $p_\psi(z)$ to match $\mathcal{M}$ topologically; see the bottom-middle panel of Figure 1 for an example.

In the presence of this topological mismatch, one might hope that $\mathcal{M}_\theta$ can sufficiently approximate $\mathcal{M}$ with enough capacity and training. However, Cornish et al. (2020) show that when this is possible, the bi-Lipschitz constant of $f_\theta$ will diverge to infinity, rendering $f_\theta$ either analytically non-injective or numerically unstable, and making density estimates unreliable (Behrmann et al., 2021). Accordingly, the topological woes of pushforward models cannot be "brute-forced" into submission.

Awareness of the data manifold's topology may be necessary for downstream applications such as defending against adversarial examples (Jang et al., 2020) or out-of-distribution detection (Caterini & Loaiza-Ganem, 2022). In the injective normalizing flows literature in particular, there has been interest in learning manifolds with multiple charts (Kalatzis et al., 2021; Sidheekh et al., 2022), which are certainly more expressive than using a single chart. Thus far, such approaches require ancillary models for inference, which can complicate density estimation, and must set the number of charts as a hyperparameter. Multiple charts also may not overlap perfectly, misspecifying the manifold.

## 2.2 Implicitly defined manifolds

The aforementioned limitations of pushforward models stem from the inability of smooth embeddings of $\mathbb{R}^m$ to characterize anything but the simplest of manifolds. A richer class of manifolds can be defined *implicitly*, as given by the following fact from differential geometry (Lee, 2013, pg.105):

---

[1] $\mathcal{M}_\theta$ may not formally be a manifold if $f_\theta$ is not an embedding because the resulting image can "self-intersect," but this distinction can be ignored in practice for density estimation models, as we will soon justify.

[2] In particular, $f_\theta$ becomes a left inverse of $g_\phi$, and equivalently, $g_\phi$ becomes a right inverse of $f_\theta$.

[3] A naive solution would be to increase the model's latent space dimensionality to $2m$; however, this would make the encoded data $\{g_\phi(x_i)\}$ singular in $\mathbb{R}^{2m}$, invalidating density estimates.

**The full-rank zero set theorem** Let $U \subseteq \mathbb{R}^n$ be an open subset of $\mathbb{R}^n$, and let $F : U \to \mathbb{R}^{n-m}$ be a smooth map. If the Jacobian matrix $J_F$ of $F$ has full rank on its zero set $F^{-1}(\{0\}) := \{x \in U : F(x) = 0\}$, then $F^{-1}(\{0\})$ is a properly embedded submanifold of dimension $m$ in $\mathbb{R}^n$.

In this paper, we exploit this theorem by constructing a neural network $F_\theta$ and defining a new model manifold $\mathcal{M}_\theta := F_\theta^{-1}(\{0\})$. We call $F_\theta$ the *manifold-defining function* (MDF) of $\mathcal{M}_\theta$. We refer to such manifolds as *implicitly defined* or *implicit*. These are not to be confused with the unrelated term *implicit generative model*, which has been used to describe both energy-based models (Du & Mordatch, 2019) and some types of pushforward models (Mohamed & Lakshminarayanan, 2016).

The zero sets of neural networks have been employed with great success for a special type of manifold: 3D shapes (Niemeyer & Geiger, 2021). An active subcommunity has formed around learning implicit 3D shapes with varying types of supervision, such as *a priori* shape information (Chen & Zhang, 2019; Mescheder et al., 2019; Park et al., 2019) or 2D images of the object (Niemeyer et al., 2020). For our context, Gropp et al. (2020) propose the most relevant method, which learns a coherent shape from a point cloud without supervision by regularizing gradients. We can reinterpret this as manifold learning, but it does not readily generalize to submanifolds where $n > m + 1$. Here we propose a way to fit $F_\theta$ to data manifolds of any dimension $m$ embedded in any dimension $n \geq m$.

## 2.3 ENERGY-BASED MODELS

Energy-based models (EBMs) have a long history in machine learning (LeCun et al., 2006) and physics (Gibbs, 1902), but Du & Mordatch (2019) introduced the first deep EBM for generative modelling. Notably, they use Langevin dynamics (Welling & Teh, 2011), a continuous MCMC algorithm, to generate samples. Training strategies and applications for EBMs have since become popular in the literature (Grathwohl et al., 2019; 2020). In particular, Xiao et al. (2021) model an EBM in the latent space of a VAE, but its training procedure maximizes full-dimensional likelihoods, making it unsuitable for density estimation on manifolds. Che et al. (2020) and Arbel et al. (2021) construct pushforward EBMs by using GAN discriminators to refine the generator's distribution; these models produce distributions on manifolds, but do not admit density estimates.

## 3 METHOD

### 3.1 NEURAL IMPLICIT MANIFOLDS

Let $F_\theta : \mathbb{R}^n \to \mathbb{R}^{n-m}$ be a smooth neural network with parameters $\theta$; our goal is to optimize it to become a manifold-defining function for $\mathcal{M}$, the data manifold. $F_\theta$ thus needs to meet two conditions:

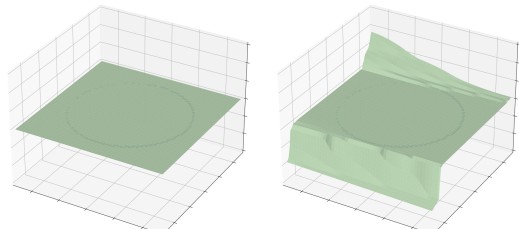

1. $F_\theta(x) = 0$ for all $x \in \mathcal{M}$.

2. $J_{F_\theta}(x)$ has full rank for all $x \in \mathcal{M}$.

Figure 2: Manifold defining functions $F_\theta$ trained without regularizing $J_{F_\theta}$. On the left, a regular neural network, has become completely flat; $F^{-1}(\{0\})$ is the entire space. On the right is the left-inverse of an injective flow, whose Jacobian has full rank analytically, but becomes numerically non-injective. These should be contrasted to the second pane of Figure 1.

Since $\mathcal{M}$ is the support of $P^*$, condition 1 is encouraged by optimizing toward the constraint $\mathbb{E}_{x \sim P^*} ||F_\theta(x)|| = 0$, which we achieve by minimizing $\mathbb{E}_{x \sim P^*} ||F_\theta(x)||^2$.

We can interpret the 3D shape objective of Gropp et al. (2020) as optimizing for condition 2 by bounding the $L_2$ norm of the gradient $J_{F_\theta}$ away from zero. However, this does not generalize to generic numbers of dimensions. Null singular values can still be present when, for example, the Frobenius norm or the operator norm of $J_{F_\theta}$ is bounded away from zero. To maintain full rank, we need to bound *all* singular values away from zero, for which we take inspiration from Kumar et al. (2020). Given their decoder $f_\theta$, they seek to make the Jacobian $J_{f_\theta}(z)$ *injective* by bounding $||J_{f_\theta}(z)u||$ away from zero for all unit-norm vectors $u \in \mathbb{R}^m$. We can do the same, except by bounding $||v^T J_{F_\theta}(x)||$ away from zero for all unit-norm $v \in \mathbb{R}^{n-m}$, since we seek to make $J_{F_\theta}(x)$

*surjective*.[4] Combining these techniques, we propose the following loss:

$$\mathcal{L}(\theta) = \mathbb{E}_{x\sim P^*, \, v\sim\text{Unif}(S)} \left[ ||F_\theta(x)||^2 + \alpha \left( \eta - ||v^T J_{F_\theta}(x)|| \right)_+^2 \right], \tag{3}$$

where $\text{Unif}(S)$ is the uniform distribution on the unit sphere $S := \{x \in \mathbb{R}^{n-m} : ||x|| = 1\}$, $(\,\cdot\,)_+$ is the ReLU function, and $\alpha$ and $\eta$ are hyperparameters determining the weight of the rank-regularization term and the minimum singular value of $J_{F_\theta}$, respectively. Empirically speaking, the Jacobian regularization term obviates degeneracy in the MDF. Without regularization, $F_\theta$ will converge towards degeneracy, even if we enforce analytical surjectivity in the Jacobian by structuring $F_\theta$ as the left-inverse of an injective flow (Kothari et al., 2021) (Figure 2, right). The unregularized flow sends $\mathbb{E}_{x\sim P^*} ||F_\theta(x)||^2 \to 0$ by bringing its singular values arbitrarily close to zero without learning the manifold. It effectively becomes numerically non-injective, akin to observed instabilities in bijective flows (Cornish et al., 2020; Behrmann et al., 2021).

**Expressivity** Making the simplifying assumption that neural networks can embody any smooth function (Hornik et al., 1989; Csáji et al., 2001), we may compare the expressivity of neural implicit manifolds with pushforward manifolds. Pushforward models can model densities on precisely those manifolds which are diffeomorphic to a subset of $\mathbb{R}^m$.

On the other hand, a broader class of manifolds can be modelled implicitly. $\mathcal{M}$ can be represented implicitly if and only if it satisfies the technical condition that its normal bundle is "trivial" (Lee, 2013, pg.271). Non-trivial normal bundles are not commonly seen in low-dimensional examples except in non-orientable manifolds such as the Möbius strip or Klein bottle. Though it is unclear whether the manifolds of most natural datasets have trivial normal bundles (eg. Carlsson et al. (2008) find a dataset of image patches to have the topology of a Klein bottle), it is certainly a much broader class than pushforward models can capture.

**Manifold arithmetic** Some datasets might satisfy multiple constraints, which one might want to learn separately before combining into a mixture or product of models. Since implicit manifold learning can be interpreted as learning a set of constraints, neural implicit manifolds exhibit composability similar to energy-based models (Hinton, 2002; Mnih & Hinton, 2005). If $F_1$ and $F_2$ are MDFs for $\mathcal{M}_1$ and $\mathcal{M}_2$ respectively, then the union $\mathcal{M}_1 \cup \mathcal{M}_2$ is the zero set of the product of functions $x \mapsto F_1(x)F_2(x)$. Concatenating outputs into the function $x \mapsto (F_1(x), F_2(x))$ instead produces the intersection $\mathcal{M}_1 \cap \mathcal{M}_2$. We note that $\mathcal{M}_1 \cup \mathcal{M}_2$ and $\mathcal{M}_1 \cap \mathcal{M}_2$ need not be manifolds anymore, meaning we can combine MDFs to form complex structures that cannot be described with a single manifold. Taking intersections and unions could, for example, be used to model conjunctions or disjunctions of data labelled with multiple overlapping attributes (Du et al., 2020).

### 3.2 Constrained Energy-Based Modelling

In this section we introduce the *constrained energy-based model* for density estimation on implicit manifolds $\mathcal{M}_\theta$. Let $E_\psi : \mathbb{R}^n \to \mathbb{R}$ be an energy function represented by a neural network and define the corresponding density as follows:

$$p_{\theta,\psi}(x) = \frac{e^{-E_\psi(x)}}{\int_{\mathcal{M}_\theta} e^{-E_\psi(y)}dy}, \quad x \in \mathcal{M}_\theta, \tag{4}$$

where $dy$ can be equivalently thought of as the Riemannian volume form or Riemannian measure of $\mathcal{M}_\theta$ (see Appendix A for details). Let $P_{\theta,\psi}$ be the resulting probability measure (we can think of $P_{\theta,\psi}$ as a probability distribution characterized by both the manifold $\mathcal{M}_\theta$ and the density $p_{\theta,\psi}$). Since the energy $E_\psi$ is defined on the full ambient space $\mathbb{R}^n$ but the corresponding model is defined only from its values on $\mathcal{M}_\theta$, we refer to $P_{\theta,\psi}$ as a *constrained energy-based model*.

Having defined $p_{\theta,\psi}$ and fixed the manifold $\mathcal{M}_\theta$, we seek to maximize log-likelihood on the data via gradient-based optimization of $E_\psi$. Since the denominator $\int_{\mathcal{M}_\theta} e^{-E_\psi(y)}dy$ is in general an intractable integral, we resort to contrastive divergence (Hinton, 2002):

$$\nabla_\psi \log p_{\theta,\psi}(x_i) = -\nabla_\psi E_\psi(x_i) + \mathbb{E}_{x\sim P_{\theta,\psi}} \left[ \nabla_\psi E_\psi(x) \right]. \tag{5}$$

---

[4]Note we are here referring to a matrix as injective (resp. surjective) if it has full column (resp. row) rank.

Importantly, the right-most term in Equation 5 is an expectation taken over $P_{\theta,\psi}$, so samples from the model are required for optimization.

**Constrained Langevin Monte Carlo**   How can one sample from $P_{\theta,\psi}$? Du & Mordatch (2019) use Langevin dynamics, a continuous MCMC method, to sample from deep EBMs. For constrained EBMs, standard Langevin dynamics is insufficient, as it will produce off-manifold samples from the energy. We need a manifold-aware MCMC method.

One such method is constrained Hamiltonian Monte Carlo (CHMC), a family of Markov chain Monte Carlo models for implicitly defined manifolds proposed by Brubaker et al. (2012). Our main contribution in this section, aside from defining constrained EBMs, is to show that CHMC – which is typically applied to analytically known manifolds – can be adapted to manifolds implicitly defined by neural networks. In particular, we show how to avoid the unstable and sometimes memory-prohibitive operation of explicitly constructing the Jacobian of $F_\theta$, which features prominently in CHMC.

We focus on the special case of constrained Langevin Monte Carlo. Fixing a step size $\varepsilon$ and omitting parameter subscripts for brevity, one iteration from position $x^{(t)}$ to $x^{(t+1)}$ requires two steps:

1. Sample a momentum $r \sim N(0, I_n)$ conditioned on membership of the tangent space of $\mathcal{M}$ at $x^{(t)}$. This can be done by sampling $r' \sim N(0, I_n)$ and projecting onto the null space of $J_F(x^{(t)})$ (written as $J_F$ for clarity):

$$r := r' - J_F^T (J_F J_F^T)^{-1} J_F r'. \tag{6}$$

2. Solve for the new position $x^{(t+1)}$ using a constrained Leapfrog step, which entails solving the following system of equations for $x^{(t+1)}$ and the Lagrange multiplier $\lambda \in \mathbb{R}^{n-m}$:

$$x^{(t+1)} = x^{(t)} + \varepsilon r - \frac{\varepsilon^2}{2} \nabla_x E(x^{(t)}) - \frac{\varepsilon^2}{2} J_F(x^{(t)})^T \lambda \tag{7}$$

$$F(x^{(t+1)}) = 0. \tag{8}$$

Now we describe how Equations 6 and 7 can be computed without constructing $J_{F_\theta}$. With access to efficient Jacobian-vector product and vector-Jacobian product routines, such as those available in functorch (He & Zou, 2021), any expression in the form of $J_F w$ for $w \in \mathbb{R}^n$ or $J_F^T v = (v^T J_F)^T$ for $v \in \mathbb{R}^{n-m}$ is tractable. Furthermore, the inverse term on the right-hand side of Equation 6 can be computed with inspiration from work in injective flows by Caterini et al. (2021) who overcome a similar expression using the conjugate gradients (CG) routine (Nocedal & Wright, 2006; Gardner et al., 2018; Potapczynski et al., 2021) and their *forward-backward auto-differentiation trick*. CG allows us to compute expressions of the form $A^{-1}b$, where $A$ is an $(n-m) \times (n-m)$ matrix. In particular, CG requires access only to the operation $v \mapsto Av$, not the matrix $A$ itself. In our case, $b = J_F r'$, a Jacobian-vector product, and the operation is $v \mapsto J_F J_F^T v$, which is again computable as a vector-Jacobian product followed by a Jacobian-vector product. Due to the shape of $J_F$, this operation is most efficiently performed using backward-mode followed by forward-mode auto-differentiation, so our method can be termed the *backward-forward* variant.

Equations 7 and 8 can be combined into a single minimization problem which can be easily optimized by second-order methods such as L-BFGS (Byrd et al., 1995):

$$\lambda^* = \arg\min_\lambda \left\| F_\theta \left( x^{(t)} + \varepsilon r - \frac{\varepsilon^2}{2} \nabla_x E_\psi(x^{(t)}) - \frac{\varepsilon^2}{2} J_{F_\theta}(x^{(t)})^T \lambda \right) \right\|, \tag{9}$$

where, in computationally challenging contexts, we can settle for suboptimal solutions at the cost of introducing bias. We note that L-BFGS outperformed first-order methods like stochastic gradient descent (Robbins & Monro, 1951) or Adam (Kingma & Ba, 2014) for this task. Once obtained, $\lambda^*$ can be plugged back into Equation 7 to directly calculate $x^{(t+1)}$.

The two steps described above constitute a single iteration of constrained Langevin dynamics. In practice, many iterations are required to obtain a good approximation to sampling from $P_{\theta,\psi}$ (Algorithm 1). Following Du & Mordatch (2019), we use a sample buffer for 95% of generated samples to assist convergence during training. To obtain completely new samples, we sample random noise in ambient space and project them to $\mathcal{M}_\theta$ by computing $\arg\min_x \|F_\theta(x)\|^2$ with L-BFGS.

---

**Algorithm 1** Constrained Langevin Monte Carlo

---

**Require:** manifold-defining function $F_\theta$, energy $E_\psi$, step size $\varepsilon$, step count $k$, initial point $x_0$

$\quad x \leftarrow x_0$
$\quad$ **for** $t = 1, \ldots, k$ **do**
$\quad\quad r' \sim N(0, I_n)$
$\quad\quad r \leftarrow r' - J_{F_\theta}^T (J_{F_\theta} J_{F_\theta}^T)^{-1} J_{F_\theta} r'$
$\quad\quad \lambda^* \leftarrow \arg\min_\lambda \left\| F_\theta \left( x + \varepsilon r - \frac{\varepsilon^2}{2} \nabla_x E_\psi(x) - \frac{\varepsilon^2}{2} J_{F_\theta}(x)^T \lambda \right) \right\|$
$\quad\quad x \leftarrow x + \varepsilon r - \frac{\varepsilon^2}{2} \nabla_x E_\psi(x) - \frac{\varepsilon^2}{2} J_{F_\theta}(x)^T \lambda^*$
$\quad$ **end for**
$\quad$ **return** $x$

---

## 4 EXPERIMENTS

The current literature on density estimation for non-trivial topologies assumes the manifold is known beforehand (Gemici et al., 2016; Mathieu & Nickel, 2020; Rezende et al., 2020; De Bortoli et al., 2022). Here we show that constrained EBMs are the best choice for such distributions in the absence of *a priori* knowledge of the manifold. We reiterate that all manifolds learned in these experiments are determined only based on samples, without additional knowledge. Quantitative comparisons of density estimates are challenging when manifolds are unknown: likelihood values are incomparable for different learned manifolds. Fortunately, we can compare the following manifolds visually.

As discussed in Section 2, the class of pushforward density estimation models is large, and any can serve as a basis of comparison. We focus on the most comparable baseline: a simple pushforward EBM consisting of an autoencoder with an EBM in the latent space. We experimented with regularizing the autoencoder by training with a Gaussian VAE objective, but it did not learn the manifold as well as a regular autoencoder (Appendix B, Figure 8). Likewise, one could replace the latent EBM with any density estimator (such as a normalizing flow (Brehmer & Cranmer, 2020) or VAE (Dai & Wipf, 2019)), but this would not affect the learned manifold.

Our code is written in PyTorch (Paszke et al., 2019). We use GPyTorch (Gardner et al., 2018) for conjugate gradients and the marching cubes algorithm of Yatagawa (2021) to plot 2D implicit manifolds in 3D. We generate synthetic data with Pyro (Bingham et al., 2019). Hyperparameter settings and other details can be found in Appendix B.

### 4.1 SYNTHETIC DATA

**Density estimation**   In our first experiment, we evaluate density estimation ability on 1000 points sampled from a mixture of two von Mises distributions on circles embedded in 2D. Results for an ordinary EBM, a pushforward EBM, and a constrained EBM are visible in Figure 3. Of note is the topology of the density learned by the pushforward EBM; it is necessarily connected and appears to be diffeomorphic to the real line except at two points of self-intersection. The constrained EBM, in contrast, captures the manifold even in regions of sparsity. The ordinary EBM is not subject to the *topological* limitations of the pushforward EBM, but it still lacks the inductive bias to learn the low intrinsic dimension of the data.

**Manifold arithmetic**   In this experiment, we highlight the ability of constrained EBMs to perform manifold arithmetic. Practical applications of this capability are left to future research. Figure 4 depicts two modes of composition for constrained EBMs. The constrained EBM depicted on the left is learned from 1000 points sampled from a balanced mixture of two projected normal distributions. After this, with no additional training, we manipulate it to create new probability models. First, two copies of the learned model are translated by 0.5 units in opposite directions.

- A new model given by the union of these two copies is depicted in the middle pane of Figure 4: it consists of the product of their MDFs and a balanced mixture of their corresponding energies. Note that the new surface self-intersects, and is no longer formally an embedded submanifold.

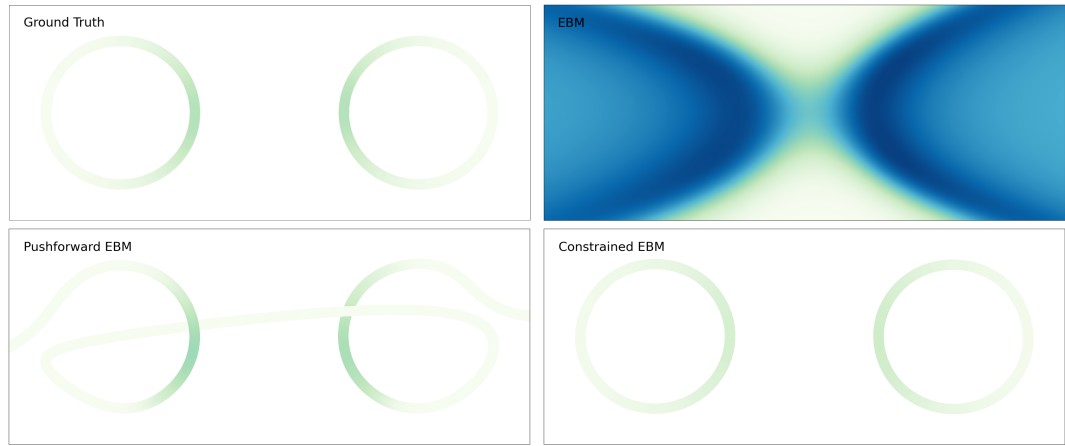

Figure 3: Manifold learning and density estimation results on a balanced, disjoint mixture of two von Mises distributions. Four models are depicted: the ground truth, an ambient EBM, a pushforward EBM, and a constrained EBM (ours).

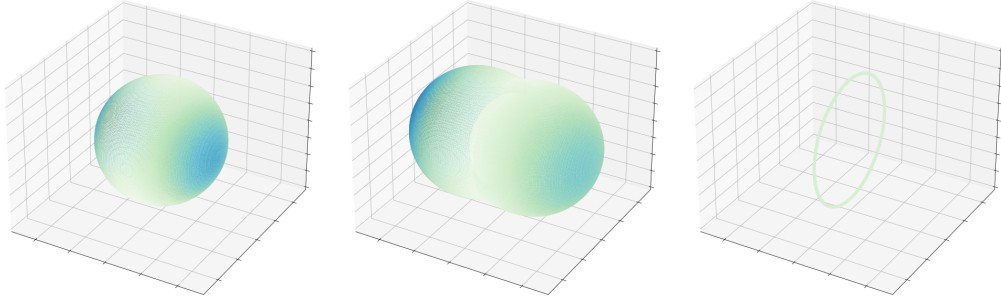

Figure 4: Manifold arithmetic with an implicitly learned sphere. From left to right: a spherical distribution learned by a constrained EBM; the union of two copies of the same model translated in different directions; and the intersection of the same two copies.

- Another new model given by the intersection of these two copies is visible in the final pane. By concatenating the output of the MDFs and summing the corresponding energies, we arrive at a circle embedded in three dimensions.

## 4.2 NATURAL DATA

**Geospatial data**    Following Mathieu & Nickel (2020), we model a dataset of global flood events from the Dartmouth Flood Observatory (Brakenridge, 2010), embedded on a sphere representing the Earth. Despite the relative sparsity of floods compared to previous datasets (they only occur on land), the constrained EBM still perfectly learns the spherical shape of the Earth (Figure 5). The pushforward EBM represents the densities fairly well, but struggles to learn the sphere and places some density off of the true manifold. Note that the constrained and pushforward EBMs are plotted using a triangular mesh and mesh grid, respectively, due to the difference in how they are defined.

**Amino acid modelling**    The structure of amino acids can be characterized by a pair of dihedral angles and thus possesses toroidal geometry. Designing flexible probabilistic models for torus-supported data is consequently of interest in the bioinformatics literature on protein structure prediction (Singh et al., 2002; Mardia et al., 2007; Ameijeiras-Alonso & Ley, 2022), and so amino acid angle data is a practical candidate for evaluating the density estimation ability of constrained EBMs. In Figure 6, we compare a constrained EBM against a pushforward EBM using an open-source amino acid dataset available from the NumPyro software package (Phan et al., 2019). Remarkably, our manifold-defining function learns the torus well in the presence of sparse data. We postulate this is because the torus is the simplest manifold matching the data's curvature. On the other hand, the pushforward EBM was unable to reliably model the manifolds.

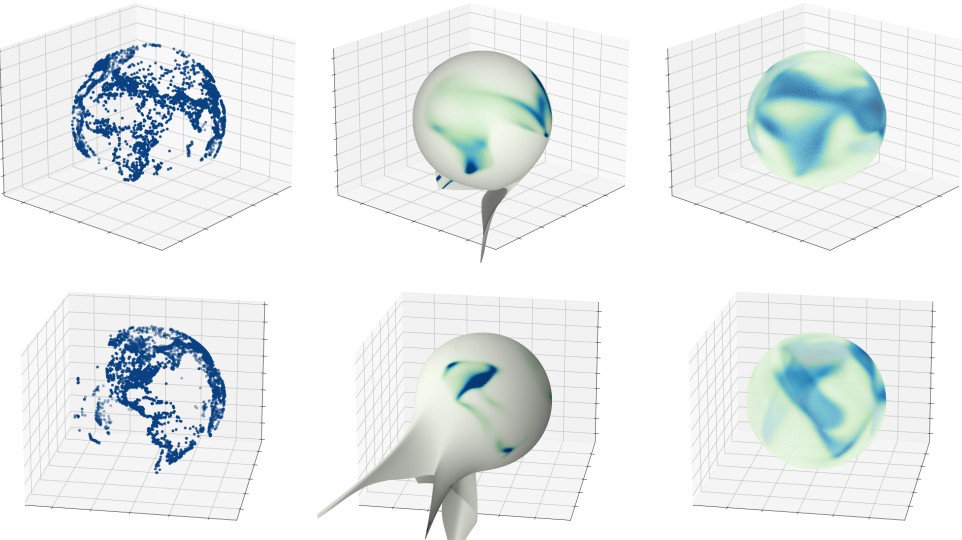

Figure 5: Manifold learning and density estimation results on flood location data. From left to right with two different viewpoints (top and bottom): the ground truth data; a pushforward EBM; and a constrained EBM (ours).

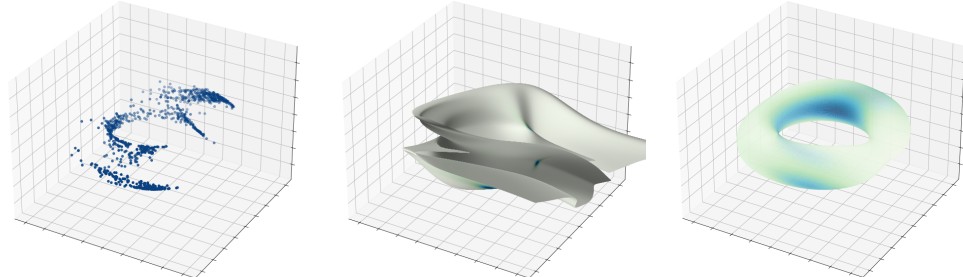

Figure 6: Manifold learning and density estimation results on the glycine angle data. From left to right: the ground truth data; a pushforward EBM; and a constrained EBM (ours).

## 5 CONCLUSION

In this paper we observed that all existing techniques to jointly learn data manifolds and densities can be described as *pushforward models*. These models must become near-diffeomorphisms, an overly strong topological limitation, in order to provide reliable density estimates. To avoid this limitation, we proposed to learn the data manifold *implicitly* with a neural network $F_\theta$. We then proposed the *constrained EBM*, a new type of EBM for modelling data on neural implicit manifolds. In both cases, we showed how the computation of the Jacobian of $F_\theta$ can be "tamed" using stochastic estimates and automatic differentiation tricks inspired by the injective flows literature (Kumar et al., 2020; Caterini et al., 2021) which frequently grapples with non-square Jacobians. Finally, we demonstrated the qualitative efficacy of constrained EBMs on both synthetic and real-world tasks.

Although we have covered the limitations of pushforward models when used for density estimation, we highlight here some of their advantages over our model. Primarily, pushforward models come with latent representations of data, which have myriad uses such as explainability and artificial reasoning (Higgins et al., 2016; Mathieu et al., 2019) and efficient density estimation in the latent space. A promising direction for future work is to combine these benefits with those of constrained EBMs.

Our model inherits all the difficulties of training EBMs; for example, it relies on the assumption that Langevin dynamics converges, which occurs only with infinite steps. Sampling remains slower than normal EBMs due to the complexity of constrained Langevin dynamics. Constrained EBMs might thus benefit from training methods that do not involve sampling, such as the Stein discrepancy (Grathwohl et al., 2020) or score-matching (Song & Kingma, 2021; De Bortoli et al., 2022).

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

# A   FORMAL SETTING

Here we expand on the formal setting in which we seek to perform density estimation.

**Geometry**   Let $\mathcal{M}$ be an $m$-dimensional Riemannian submanifold of ambient space $\mathbb{R}^n$ where $m < n$. Formally this refers to the pair $(\mathcal{M}, \mathbf{g})$, where $\mathcal{M} \subseteq \mathbb{R}^n$ is a manifold and $\mathbf{g}$ is the Riemannian metric inherited from ambient Euclidean space. In other words, $\mathbf{g}$ is the restriction of the canonical Euclidean metric, which is characterized by the standard dot product between vectors, to vectors which are tangent to $\mathcal{M}$. The metric $\mathbf{g}$, which is typically implied, captures the curvature information we would like to associate with $\mathcal{M}$.

A manifold's Riemannian metric gives rise to a unique differential form known as the Riemannian volume form $d\mu$, which allows for the integration of continuous, compactly supported, real-valued functions $h$ over the Riemannian manifold (Lee, 2013):

$$\int_{\mathcal{M}} h \, d\mu. \tag{10}$$

**Probability**   Let $\{x_i\}$ be observed samples drawn from $P^*$, a probability measure supported on $\mathcal{M}$. Since $\mathcal{M}$ has a lower intrinsic dimension than $\mathbb{R}^n$, it is "infinitely thin." In other words, $P^*(\mathcal{M}) = 1$ while the (Lebesgue) volume of $\mathcal{M}$ is 0, meaning no probability density integrated over the ambient space can be used to represent $P^*$. Formally stated, $P^*$ is not absolutely continuous with respect to the Lebesgue measure on $\mathbb{R}^n$.

Instead, we require a new way to define the volumes of subsets of $\mathcal{M}$. We can then formally define a probability density $p^*$ over $\mathcal{M}$ and integrate with respect to this volume to obtain probabilities. The volume form $d\mu$ on $\mathcal{M}$ is the answer; the probability of a set $S \subseteq \mathcal{M}$ can be computed as follows:

$$P^*(S) = \int_S p^* \, d\mu. \tag{11}$$

We note that the volume form $d\mu$ from differential geometry is not technically a measure in the sense of measure theory. This obstacle is minor: $d\mu$ can be extended to a true measure by a common measure-theoretic tool known as the Riesz-Markov-Kakutani representation theorem[5] (Rudin, 1987). Thus we may identify $d\mu$ with a measure $\mu$ on $\mathcal{M}$ which produces volumes of Borel sets in $\mathcal{M}$ and which we call the Riemannian measure of $\mathcal{M}$ (Pennec, 1999).

Formally, we require $P^*$ to be absolutely continuous with respect to $\mu$, and we thus write that $p^*$ is the Radon-Nikodym derivative of $P^*$ with respect to $\mu$: $p^* = \frac{dP^*}{d\mu}$. This is the ground-truth density function we seek to model in this work.

# B   EXPERIMENT DETAILS

For all experiments, we use feedforward networks with SiLU activations (Hendrycks & Gimpel, 2016; Ramachandran et al., 2017). All models are trained with the Adam optimizer with the default PyTorch parameters, except for the learning rate which is set as described below (Kingma & Ba, 2014). All EBMs, constrained EBMs, and pushforward EBMs are trained with a buffer size of 1000, from which we initialize each Langevin dynamics sample with 95% probability. We do not use spectral normalization for EBMs: we found it harmed the quality of density estimates. Initial noise for the constrained EBM is sampled uniformly from a box in ambient space containing the ground truth manifold and then projected to the manifold by solving for $\arg\min_{x_{\text{noise}}} ||F_\theta(x_{\text{noise}}))||^2$ with L-BFGS using strong Wolfe line search. Equation 9 is also optimized using a single step of L-BFGS with strong Wolfe line search. All models were tuned by hand for visual performance. Training times are reported below, but we caution that models were not tuned for runtime, so the raw times should not be compared between models to evaluate efficiency.

To plot the constrained EBM densities, we estimate the normalizing constants using Monte Carlo. Since the learned MDFs always provide very good approximations of the true manifolds, we estimate

---

[5]In the reference and sometimes in general, this theorem is called the Riesz representation theorem, which can also refer to a different theorem about Hilbert spaces.

each normalizing constant using uniform samples from the *ground truth* manifold for convenience. To plot the pushforward EBM densities, we estimate the normalizing constants in latent space with Monte Carlo estimates based on uniform sampling within the clamped bounds. We then compute pushforward densities with Equation 2.

All experiments were performed on an Intel Xeon Silver 4114 CPU.

We provide quantitative results in Table 1. We estimate the distance of each training point to the manifold using an optimization procedure, and report minimum, median, mean, and maximum distances over the training set. Nearest-point estimates must be computed differently for constrained EBMs and pushforward EBMs, and therefore estimates for each model are prone to different sources of error, so these metrics should be used only with caution as a basis of comparison. For the constrained EBM with MDF $F_\theta$, we compute the nearest point on the manifold to datapoint $x_i$ as

$$x^* = \arg\min_x ||x - x_i||^2 + 10^{10}||F_\theta(x)||^2,$$

where $x$ has been initialized to $x_i$. For the pushforward EBM with encoder-decoder pair $(f_\theta, g_\phi)$, we compute the nearest point as $f_\theta(z^*)$, where

$$z^* = \arg\min_z ||f_\theta(z) - x_i||^2,$$

where $z$ has been initialized to $g_\phi(x_i)$. On every dataset, the implicit manifold captures the training set with substantially more accuracy.

Table 1: Statistics of estimated dataset distances to the manifold.

| EXPERIMENT | CONSTRAINED EBM | | | | PUSHFORWARD EBM | | | |
|---|---|---|---|---|---|---|---|---|
| | MIN | MEDIAN | MEAN | MAX | MIN | MEDIAN | MEAN | MAX |
| MOTIVATING EXAMPLE | $0.000 \times 10^{-5}$ | $0.07 \times 10^{-2}$ | $0.08 \times 10^{-2}$ | 0.006 | $0.226 \times 10^{-5}$ | $0.13 \times 10^{-2}$ | $0.21 \times 10^{-2}$ | 0.184 |
| DENSITY ESTIMATION | $0.639 \times 10^{-5}$ | $0.34 \times 10^{-2}$ | $0.36 \times 10^{-2}$ | 0.013 | $1.741 \times 10^{-5}$ | $0.53 \times 10^{-2}$ | $0.65 \times 10^{-2}$ | 0.072 |
| MANIFOLD ARITHMETIC | $1.016 \times 10^{-5}$ | $0.19 \times 10^{-2}$ | $0.20 \times 10^{-2}$ | 0.006 | - | - | - | - |
| GEOSPATIAL DATA | $0.156 \times 10^{-5}$ | $0.06 \times 10^{-2}$ | $0.08 \times 10^{-2}$ | 0.004 | $0.780 \times 10^{-5}$ | $0.13 \times 10^{-2}$ | $0.16 \times 10^{-2}$ | 0.014 |
| AMINO ACID MODELLING | $0.365 \times 10^{-5}$ | $0.66 \times 10^{-2}$ | $0.77 \times 10^{-2}$ | 0.045 | $4.295 \times 10^{-5}$ | $2.42 \times 10^{-2}$ | $3.00 \times 10^{-2}$ | 0.627 |

## B.1 SYNTHETIC DATA

**Motivating example (Figure 1)** We sampled 1000 points from a von Mises distribution on a unit circle centred at $(0, 0)$ with the mode located at $(1, 0)$ and a concentration of 2.

The MDF for the constrained EBM consisted of 3 hidden layers with 8 units per hidden layer. The MDF was trained for 200 epochs with a batch size of 100, a learning rate of 0.01, $\eta = 1$, and $\alpha = 1$. Training took 6.56 seconds.

The energy function for the constrained EBM consisted of 2 hidden layers with 32 units per hidden layer. It was trained for 40 epochs with a batch size of 100, a learning rate of 0.01, gradients clipped to a norm of 1, and energy magnitudes regularized with a coefficient of 0.1. Langevin dynamics at each training step were run for 10 steps with $\varepsilon = 0.3$, a step size of 1, and energy gradients clamped to maximum values of 0.1 at each step. Training took 4 minutes, 5 seconds. In Figure 7, we evaluate the effect of the Langevin dynamics step count on training dynamics, where we vary the step size (and remove the training buffer, as this effectively increases the average step count). Fewer steps leads to a more peaked mode because the estimated model distribution is overly smooth when estimating the right-hand side of Equation 5.

The pushforward EBM's encoder and decoder each had 3 hidden layers with 32 units per hidden layer. They were jointly trained for 300 epochs with a batch size of 100, a learning rate of 0.001, and gradients clipped to a norm of 1. Training took 16.4 seconds.

The pushforward EBM's energy function had 3 hidden layers and 32 units per hidden layer. It was trained for 200 epochs with a batch size of 100, a learning rate of 0.01, gradients clipped to a norm of 1, and energy magnitudes regularized with a coefficient of 0.1. Langevin dynamics at each training step were run for 60 steps with $\varepsilon = 0.5$, a step size of 10, and energy gradients clamped to maximum values of 0.03 at each step. Training took 5 minutes, 19 seconds.

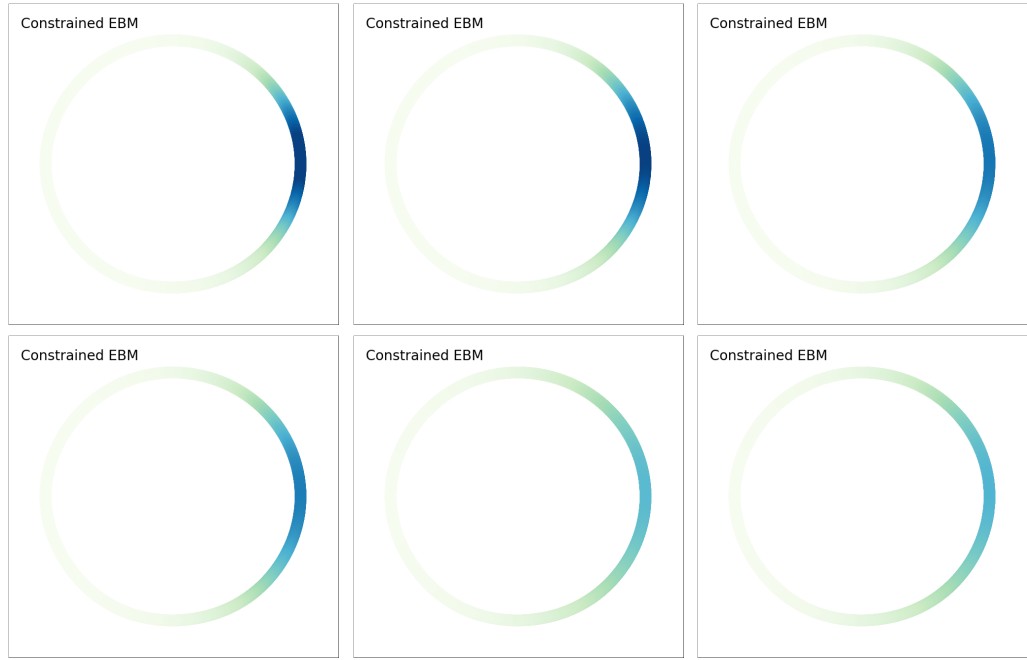

Figure 7: Constrained EBM manifold learning and density estimation results on a von Mises distribution where Langevin dynamics during training has been run (with no replay buffer) with different step counts. From left to right and top to bottom, step counts per training step: 1, 3, 5, 10, 20, 40. The setting of 20 langevin dynamics steps is sufficient for convergence.

**Density estimation**  We sampled 1000 points from a balanced mixture of two von Mises distributions with concentration 2 on circles of unit radius. Respectively, they are centred at $(-2, 0)$ and $(2, 0)$ with modes at $(-1, 0)$ and $(1, 0)$ (or, at polar angles of 0 and $\pi$ with respect to the centre of each circle).

The MDF for the constrained EBM consisted of 3 hidden layers with 8 units per hidden layer. The MDF was trained for 1000 epochs with a batch size of 100, a learning rate of 0.01, $\eta = 1$, and $\alpha = 1$. Training took 31.3 seconds.

The energy function for the constrained EBM consisted of 3 hidden layers with 32 units per hidden layer. It was trained for 50 epochs with a batch size of 100, a learning rate of 0.01, gradients clipped to a norm of 1, and energy magnitudes regularized with a coefficient of 0.3. Langevin dynamics at each training step were run for 10 steps with $\varepsilon = 0.4$, a step size of 1, and energy gradients clamped to maximum values of 0.1 at each step. Training took 37.8 seconds.

The (ambient) EBM consisted of 2 hidden layers with 32 units per hidden layer (we found that using only 2 hidden layers gave a smoother density). It was trained for 3 cycles of 200 epochs with a step size of 10. We used a batch size of 100, a learning rate of 0.01, gradients clipped to a norm of 1, and energy magnitudes regularized with a coefficient of 0.5. Langevin dynamics at each training step were run for 10 steps with $\varepsilon = 0.1$ and energy gradients clamped to maximum values of 0.03 at each step. Training took 3 minutes, 2 seconds.

The pushforward EBM's encoder and decoder each had 3 hidden layers with 32 units per hidden layer. It was trained for 1000 epochs with a batch size of 100, a learning rate of 0.001, and gradients clipped to a norm of 1. Training took 57.3 seconds. We also tried training the autoencoder using a variational autoencoder loss, but found that to learn the manifold properly, the KL term had to be heavily downweighted near the point of nonexistence. In Figure 8 we show how manifold learning ability deteriorates as the KL-weighting is increased.

The pushforward EBM's energy function had 3 hidden layers and 32 units per hidden layer. It was trained for 300 epochs with a batch size of 100, a learning rate of 0.01, gradients clipped to a norm of

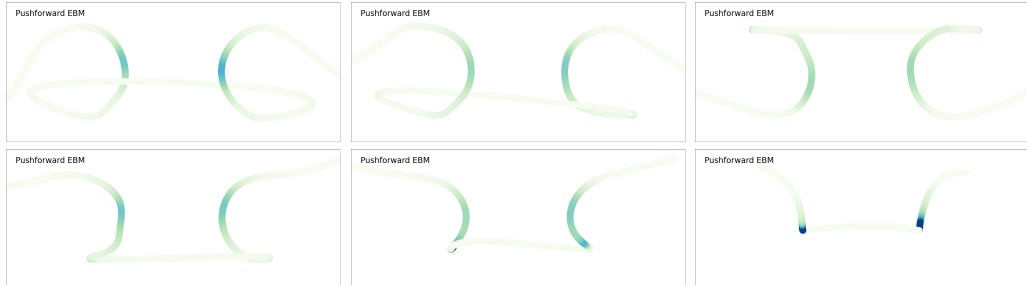

Figure 8: Manifold learning and density estimation performance for different weightings $\beta$ on the KL-divergence term of the VAE loss. From left to right, top to bottom: $\beta = 0.01, \beta = 0.03, \beta = 0.05, \beta = 0.1, \beta = 0.2, \beta = 0.4$.

1, and energy magnitudes regularized with a coefficient of 0.1. Langevin dynamics at each training step were run for 60 steps with $\varepsilon = 1.0$, a step size of 10, and energy gradients clamped to maximum values of 0.03 at each step. Training took 7 minutes, 5 seconds.

**Manifold arithmetic**   We sampled 1000 points from a balanced mixture of two projected normal distributions on the unit sphere. Each component was a normal distribution with unit diagonal covariance centred at $(1, 0, 0)$ and $(-1, 0, 0)$ respectively before being projected to the sphere.

The MDF for the constrained EBM consisted of 3 hidden layers with 8 units per hidden layer. The MDF was trained for 1500 epochs with a batch size of 100, a learning rate of 0.01, $\eta = 1$, and $\alpha = 2$. Training took 46.9 seconds.

The energy function for the constrained EBM consisted of 2 hidden layers with 32 units per hidden layer. It was trained for 5 rounds of 10 epochs each wherein Langevin dynamics was run for 5, 10, 20, 40, and 50 steps respectively. We used a batch size of 50, a learning rate of 0.01, gradients clipped to a norm of 1, and energy magnitudes regularized with a coefficient of 1. Langevin dynamics at each training step were run for 10 steps with $\varepsilon = 0.1$, a step size of $\varepsilon^2$, and energy gradients clamped to maximum values of 0.03 at each step. Training took 17 minutes, 51 seconds.

## B.2   NATURAL DATA

**Geospatial data**   We modelled floods from the Dartmouth Flood Observatory's global active archive, which is available without charge for research and education purposes.

The MDF for the constrained EBM consisted of 3 hidden layers with 8 units per hidden layer. The MDF was trained for 500 epochs with a batch size of 100, a learning rate of 0.01, $\eta = 1$, and $\alpha = 2$. Training took 1 minute, 11 seconds.

The energy function for the constrained EBM consisted of 4 hidden layers with 32 units per hidden layer. It was trained for 4 rounds of 10 epochs each wherein Langevin dynamics was run for 5, 10, 20, and 40 steps respectively. We used a batch size of 100, a learning rate of 0.01, gradients clipped to a norm of 1, and energy magnitudes regularized with a coefficient of 1. Langevin dynamics at each training step was run with $\varepsilon = 0.1$, a step size of $\varepsilon^2$, and energy gradients clamped to maximum values of 0.03 at each step. Training took 27 minutes, 43 seconds.

The pushforward EBM's encoder and decoder each had 4 hidden layers with 32 units per hidden layer. They were jointly trained for 500 epochs with a batch size of 100, a learning rate of 0.001, and gradients clipped to a norm of 1. Training took 2 minutes, 29 seconds.

The pushforward EBM's energy function had 4 hidden layers and 32 units per hidden layer. It was trained for 50 epochs with a batch size of 100, a learning rate of 0.01, gradients clipped to a norm of 1, and energy magnitudes regularized with a coefficient of 0.1. Langevin dynamics at each training step were run for 60 steps with $\varepsilon = 0.5$, a step size of 10, and energy gradients clamped to maximum values of 0.03 at each step. Training took 4 minutes, 42 seconds.

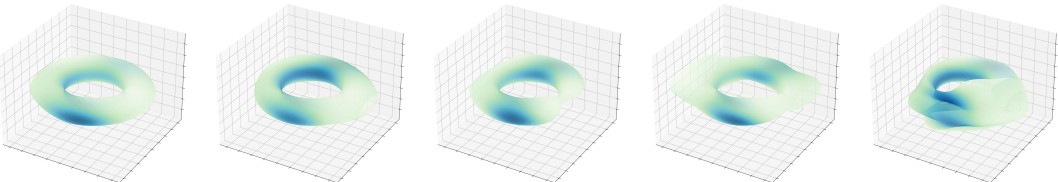

Figure 9: Constrained EBM manifold learning and density estimation results on the glycine angle data for different values of $\eta$, the hyperparameter setting the boundary under which singular values will be penalized by the Jacobian regularization term. From left to right: $\eta = 0.3$, $\eta = 1$, $\eta = 2$, $\eta = 3$, and $\eta = 5$.

**Amino acid modelling**  The MDF for the constrained EBM consisted of 2 hidden layers with 8 units per hidden layer. The MDF was trained for 500 epochs with a batch size of 100, a learning rate of 0.01, $\eta = 0.3$, and $\alpha = 1$. We found that increasing $\eta$, the smallest singular value required of $J_{F_\theta}$ by the regularization term, made the implicit manifold harder to optimize. This occasionally yielded plateaus in the loss function and resulted in incorrect manifolds, depicted in Figure 9. Training took 13.6 seconds.

The energy function for the constrained EBM consisted of 2 hidden layers with 32 units per hidden layer. It was trained for 2 rounds of 10 epochs each wherein Langevin dynamics was run for 5 and 10 steps respectively. We used a batch size of 100, a learning rate of 0.01, gradients clipped to a norm of 1, and energy magnitudes regularized with a coefficient of 1. Langevin dynamics at each training step was run for 10 steps with $\varepsilon = 0.1$, a step size of $\varepsilon^2$, and energy gradients clamped to maximum values of 0.03 at each step. Training took 1 minute, 14 seconds.

The pushforward EBM's encoder and decoder each had 4 hidden layers with 32 units per hidden layer. They were jointly trained for 500 epochs with a batch size of 100, a learning rate of 0.001, and gradients clipped to a norm of 1. Training took 33.5 seconds.

The pushforward EBM's energy function had 3 hidden layers and 32 units per hidden layer. It was trained for 50 epochs with a batch size of 100, a learning rate of 0.01, gradients clipped to a norm of 1, and energy magnitudes regularized with a coefficient of 0.1. Langevin dynamics at each training step were run for 60 steps with $\varepsilon = 0.5$, a step size of 10, and energy gradients clamped to maximum values of 0.03 at each step. Training took 54 seconds.

