# OpenReview forum: "Neural Implicit Manifold Learning for Topology-Aware Generative Modelling"
_ICLR.cc/2023/Conference — Submitted to ICLR 2023_

### Official Review · Reviewer_sPaT · 2022-10-20

**Confidence:** 2
**Correctness:** 4
**Technical Novelty And Significance:** 3
**Empirical Novelty And Significance:** 2
**Recommendation:** 6

**Clarity, Quality, Novelty And Reproducibility:**

It is true that embedding an m-dimensional manifold into an m-dimensional euclidean space is challenging. However, in many applications, people tend to use much higher dimensions. For example, in the flow based density estimation, people use the dimension of the ambient space n instead of the dimension of the manifold m in the latent density estimation. Frankly, in real applications, it is hard to estimate the dimension m of the manifold. Instead the dimension of the latent space is chosen empirically. I conjecture in most of those applications the dimension used in the latent space is way higher than the “real” dimension of the manifold. If this is the case, the impact of this paper will be quite limited.

**Strength And Weaknesses:**

Strength: The challenge the authors raised was real and interesting. I feel not many people have realized that. The proposed implicit model, as well the whole procedure of optimization, are novel.
Weakness: The empirical part of the paper is weak. Only synthetic datasets were used. If the proposed model is so powerful, why not use it on real-world datasets?


**Summary Of The Paper:**

This paper points out some potential issues and limitations using “pushforward” models for density estimation. The main issue is that the general manifold of dimension m may not be effectively embedded in the latent space of the same dimension m. To resolve that challenge, the authors propose to construct an implicit model manifold, called MDF, so that the original data manifold lives on the null set of the MDF. The new implicit model covers a richer class of manifolds. Following this line, the authors further propose to use an energy based model to model the density over the manifold. Also the constrained Langevin Monte Carlois used to sample from the manifold.


**Summary Of The Review:**

Overall my review is mixed: on one hand I do like the issues/challenges raised by the authors. Also the proposed solution seems very novel. On the other hand, I feel the challenges raised may not be real in applications since people are using dimensions much higher than m in the latent space. Also there are no real-world datasets used in the experimental sections. This makes me question whether the proposed novel but complicated solution is truly better than existing “pushforward” models.

---

> ### Author Response · Authors · 2022-11-11
> **Response to Reviewer sPaT**
>
> Thank you for the thoughtful review. We respond to your concerns about synthetic data in the general response above.
>
> We agree with your concern that determining the dimension of the unknown data manifold is challenging, but this does not mean we should give up on building well-specified probabilistic models for such datasets. We can start by applying techniques from the dimensionality estimation literature (eg. [A]), but we agree that this problem is far from solved.
>
>
> [A] Levina, Elizaveta, and Peter Bickel. "Maximum likelihood estimation of intrinsic dimension." Advances in neural information processing systems 17 (2004).

---

### Official Review · Reviewer_VPPU · 2022-11-02

**Confidence:** 4
**Correctness:** 4
**Technical Novelty And Significance:** 3
**Empirical Novelty And Significance:** 2
**Recommendation:** 3

**Clarity, Quality, Novelty And Reproducibility:**

Clarity
-------
The paper is quite clear to read. However, the section on energy-based modeling felt rushed (not much of the background/methods are given), making it hard to parse for someone who is not particularly familiar with that literature. The CHMC section's clarity could be improved with better writing (ie an introduction of the method first, followed by places where the randomized numerical linear algebra techniques could be applied).

Quality
--------
The overall quality is reasonably high. The concepts that are developed are reasonably intensive, and the method seems like a good bridge between existing literatures.

Novelty
---------
The method is marginally novel. It mostly applies existing techniques to an existing problem, although it does generalize the existing techniques somewhat.

Reproducibility
-----------------
The code is given in the supplementary zip, so it is reasonably reproducible.

**Strength And Weaknesses:**

Strengths
-----------
* The paper brings together a lot of disparate literature to build better manifold-learning flow methods.
* The results show the first ability to learn topologically nontrivial manifold structures from data.

Weaknesses
--------------
* Outside of showing that the methods (implicits, ebms, CHMC) are of practical interest, the technical contributions feel marginal. In particular, for implicits, the geometric regularization term is similar to previous work (some mentioned in the paragraph) except as a generalization to matrices, and, for CHMC, the contribution seems like an application existing random numerical linear algebra techniques (hutchinson + CG) to the standard CHMC algorithm.
* The experimental section feels very toy. In particular, section 4.1 just shows that the method works (and validates the manifold arithmetic component). Section 4.2 is more interesting, but notably doesn't test on data that is sampled from an unknown manifold (ie it is hard to motivate the method if we know the manifold is a sphere/torus). More importantly, all of these examples are *low dimensional*. Since the developed theory is built specifically to generalize to submanifolds of arbitrary codimension (i.e. the jacobian regularization method)  or for higher dimensional spaces (i.e. the novel CHMC method is built to scale to higher dimensions), I would expect an example showing that the developed theory matters, something that isn't true for the example surfaces in R^3.
* The proposed methodology presents several important drawbacks. First, we can't evaluate log probabilities (without resorting to evaluating the energy function on all points of the manifold, which is computationally infeasible especially for dimension $> 3$). Second, the manifold and the probability density are two separate models (in particular the EBM is trained as an ambient density, not an intrinsic manifold density). Third, sampling relies on an expensive MCMC technique. Since the purpose of manifold-learning flows was to handle these issues, the current framing as a manifold-learning flow paper is somewhat strange/out-of-place.

**Summary Of The Paper:**

This paper proposes a method to learn implicit manifolds and sample from them using energy-based models. In particular, the paper first proposes a regularized method for learning submanifolds of any codimension and then shows how to fit the energy function and sample using a faster version of constrained HMC.

**Summary Of The Review:**

Overall, I vote to reject the paper. My reasoning is that, while the paper does deftly utilize preexisting methods to construct manifold-learning flows, the adaptation is somewhat straightforward (e.g. regularize all singular values + use random numerical linear algebra techniques + backprop for CHMC). Importantly, the experiments don't showcase the primary benefits of the developed techniques, calling into question said developments. Either the manifold structure is known a priori or existing methods (e.g. a combination of Gropp et al. + standard CHMC) should suffice since the manifold is a surface in R^3. Of secondary concern, I believe that the current framing as a "manifold learning flow" paper is inaccurate, as the proposed method seems to not be addressing the same issues as the field. I would instead encourage the authors to perhaps motivate it from a different lens (perhaps EBM + adding in a manifold constraint).

---

> ### Author Response · Authors · 2022-11-11
> **Response to Reviewer VPPU**
>
> We appreciate your time and constructive feedback on the clarity, quality and content of the work. You praised the paper's synthesis of disparate literature to build a density estimation method, but felt that our technical contribution was marginal on top of this literature. You also expressed some misgivings about the experiments. We address these concerns in the general response.
>
> We admit to some confusion over your claim that we are framing our work as a “manifold-learning flow paper.” Can you elaborate on what you meant by this? Our goal was to perform density estimation on manifolds, and while manifold-learning flows are one of many ways to do this, we do not use flows in our work, and we are not necessarily trying to address all the same issues as manifold-learning flows.

---

> > ### Comment · Reviewer_VPPU · 2022-11-21
> > **Manifold-Learning Flows**
> >
> > Thank you for your response. I was primarily referring to the overarching narrative of the paper, which explicitly builds around comparisons with manifold learning flows. For example, this shows up as early as the abstract: "Such procedures, which we call pushforward models, incur a straightforward limitation: [...] To remedy this problem, we propose to model M as a neural implicit manifold:".

---

> > > ### Author Response · Authors · 2022-11-21
> > > **On the Framing of Our Paper**
> > >
> > > We thank you for the clarification. We would characterize your concern as being more about our comparisons to pushforward methods in general: manifold-learning flows are but one of the many pushforward methods we discuss for manifold learning and density estimation tasks (admittedly most are not motivated as manifold learning methods the way manifold-learning flows are).
> > > Your main concern in this regard is that, despite our comparisons to them, our model lacks some benefits of pushforward models: direct density evaluation, a low-dimensional latent space for density estimation, and single-pass sampling.
> > > - We emphasize however that our motivation as outlined in the paper is not to supplant these methods and all their advantages; rather, it is to propose a density model for non-trivial differential topologies. The pushforward model framework is just the set of baselines which comes closest to doing this. This necessitates discussing them in detail. In our final version we will re-order some of our discussion to emphasize that replacing pushforward models is not our goal, though we can no longer submit revisions at present.
> > > - For clarification we point out that it is viable to estimate normalizing constants in somewhat higher dimensions, thus allowing us to report normalized log probabilities. For example, [A] uses annealed importance sampling [B] to estimate the partition function of an EBM on MNIST, which is 784-dimensional. Moreover, unnormalized log probabilities are pretty much just as useful as normalized ones for downstream tasks; eg. for out-of-distribution detection or inpainting.
> > >
> > >
> > > [A] Du, Yilun, and Igor Mordatch. “Implicit generation and modeling with energy based models.” Advances in Neural Information Processing Systems 32 (2019).
> > >
> > > [B] Neal, Radford M. “Annealed importance sampling.” Statistics and computing 11.2 (2001): 125-139.

---

### Official Review · Reviewer_aqkP · 2022-11-03

**Confidence:** 4
**Correctness:** 2
**Technical Novelty And Significance:** 2
**Empirical Novelty And Significance:** 2
**Recommendation:** 5

**Clarity, Quality, Novelty And Reproducibility:**

The quality and clarity of the writing are good.
The work is not fully original. The proposed approach combines several standard building blocks, so its novelty is limited. The code is provided, and the results seem reproducible.

**Strength And Weaknesses:**

Strength:
- a manifold learning and density estimation on the manifold is an important problem statement for many applications;
- the proposed approach is simple and easy to follow;
- the authors applied the approach to several;
- the idea of modeling a manifold as an implicit function is interesting, although it is not novel. For example, in https://arxiv.org/pdf/2011.12026.pdf (proc. of CVPR, 2021), the authors proposed to represent a manifold of images based on an implicitly defined GAN model,
- the proposed idea of manifold arithmetic sounds rather interesting, where thanks to implicit representation, we can model various operations on a given set of manifolds.

Weakness:
- the approach is a combination of several existing and relatively standard building blocks like implicit functions based on neural networks, an energy model, a Langevin dynamics;
the authors did not make any ablation study proving that the proposed selections of these building blocks and their hyperparameters are optimal,
- the proposed approach for the manifold arithmetic requires additional testing. It is not clear for which applied problems it can be used or which existing solutions it can improve,
- the manifolds considered in the experimental section are low-dimensional. Actually, they resemble some 3D shapes of visual objects. So the difference between the proposed approach and the approaches for shape modeling based on implicit functions, see, e.g., the paper https://arxiv.org/pdf/2008.06520.pdf (Learning Gradient Fields for Shape Generation) is not clear
- the authors did not test their approach on any standard manifold learning problems, even those proposed in 200x, see, e.g., https://lvdmaaten.github.io/publications/papers/TR_Dimensionality_Reduction_Review_2009.pdf

**Summary Of The Paper:**

The paper proposes an approach to estimate a manifold using an implicit model. The authors use a constrained energy-based model to learn the data distribution within the manifold. A Langevin dynamics is used to sample from the manifold while training and at an inference time.

**Summary Of The Review:**

- the idea of using implicit models for manifold representation is interesting, and I have not seen papers explicitly proposing such an idea,
- the proposed algorithm is not novel; it consists of standard building blocks,
- the authors did not conduct any ablation study to investigate the influence of different parts of the algorithm and did not perform sufficient experimental testing of the proposed approach for manifold learning.

---

> ### Author Response · Authors · 2022-11-11
> **Response to Reviewer aqkP**
>
>   We thank you for your review and extensive feedback. We appreciate your positive feedback about the clarity of the writing and importance of the problem. Our general response addresses some of your concerns about novelty and experiments, which we hope you find satisfactory. Below we address your other concerns:
>
> - You suggested that modeling a data manifold as an implicit function may not be novel, citing [A] and [B] as examples.
>
>     - We point out that [A] does *not* represent a manifold of images implicitly, as would indeed be relevant to our work. Instead, it generates implicit neural representations for images, which is common in the literature, and only related insofar as it contains the word *implicit*.
>
>     - [B] is more relevant in that they seek to generate data on manifolds, but again distinct in its aim, method, and capabilities. They use denoising score-based models to generate point clouds on surfaces - these models are already well known to approximate manifold-supported distributions well, even when their topology is complex [C, D, E].
>
>       Our chief motivation, however, is to learn probability densities of manifold-supported data. Denoising score-based models cannot do this (their density estimates ignore manifold structure). Furthermore, there is again no methodological similarity beyond the word *implicit* and mention of the broader concept of energy-based modelling, which is used in [B] to motivate denoising score matching.
>
> - You criticized the lack of ablation study on the components of the model - the implicit manifold, the EBM, and langevin dynamics. All of these components are necessary to the model and cannot be meaningfully tested except as a whole. Can you clarify what you would expect to see in an ablation study? We note that we do in fact discuss hyperparameter selection in Appendix B.
>
> - You recommended testing on some standard manifold-learning problems, but the ones provided in your review are in fact *dimensionality reduction* problems. Dimensionality reduction is not a goal or capability of our method.
>
>
> [A] Skorokhodov, Ivan, Savva Ignatyev, and Mohamed Elhoseiny. "Adversarial generation of continuous images." Proceedings of the IEEE/CVF Conference on Computer Vision and Pattern Recognition. 2021.
>
> [B] Cai, Ruojin, et al. "Learning gradient fields for shape generation." European Conference on Computer Vision. Springer, Cham, 2020.
>
> [C] Song, Yang, and Stefano Ermon. "Generative modeling by estimating gradients of the data distribution." Advances in Neural Information Processing Systems 32 (2019).
>
> [D] Pidstrigach, Jakiw. "Score-Based Generative Models Detect Manifolds." arXiv preprint arXiv:2206.01018 (2022).
>
> [E] Salmona, Antoine, et al. "Can Push-forward Generative Models Fit Multimodal Distributions?." arXiv preprint arXiv:2206.14476 (2022).

---

### Author Response · Authors · 2022-11-11
**General Response**

We thank all reviewers for the valuable time and effort they have put into the review process. We were pleased to see that the reviewers found that our work “addresses an important problem” (**aqkP**), is clear (**aqkP**, **VPPU**), interesting (**aqkP**, **sPaT**), and novel (**sPaT**). We address two common concerns here in the general response, and individual concerns in direct replies.

- Reviewers **aqkP** and **VPPU** felt that our method consisted of simple components, each from previous work, limiting its novelty.

  Our main contributions are parametrizing the data manifold as the zero set of a neural network and performing density estimation on this manifold. Neither has been done before. One could indeed argue that each building block is only slightly altered from its original source, but any research contribution viewed at a high enough resolution will look like a straightforward composition of known techniques. We believe we have tied past work together in a new way to solve a new problem: topology-aware density estimation.

- All reviewers brought up our choice of experiments as a concern. In particular, reviewers **aqkP** and **VPPU** are concerned that they are low-dimensional, and reviewer **sPaT** said that “only synthetic datasets were used.”

  We highlight that we do indeed model real-world datasets of practical interest in section 4.2. Work focusing on these low-dimensional settings (eg. Mathieu and Nickel (2020)) is already of interest in the machine learning community. The ability to model densities on them without a priori knowledge is certainly a novel contribution, whether or not the manifold loss is distinct from Gropp et al. (2020) or the sampling procedure uses autodiff tricks. These technicalities, while useful, are subordinate to our central contribution, which is the ability to learn *densities* on manifolds of nontrivial topologies.

     We also note that we have yet to achieve good results on higher-dimensional image datasets such as MNIST, although our proposed loss and speedups have allowed us to run our model on such data. We hope our work encourages the development of more performant constrained EBMs for images and other high-dimensional data.

---

### Author Response · Authors · 2022-11-16
**Discussion Ending Reminder**

We kindly remind our reviewers that the discussion period comes to an end in two days. Do you have any further concerns we can address?

---

### Decision · Program_Chairs · 2023-01-20

**Decision:**

Reject

**Justification For Why Not Higher Score:**

The paper does not meet the standards of novelty, significance, empirical evaluation, and clarity required for a high-quality publication. The paper does not make any substantial or original contribution to the field of manifold learning and density estimation, and does not provide any convincing evidence or validation for its claims and results. The paper does not clearly explain or motivate its problem and method, and does not compare or contrast its approach with other related work. The paper does not address the reviewers' concerns and criticisms adequately, and does not provide any rebuttal or revision to improve its quality and significance.

**Justification For Why Not Lower Score:**

N/A

**Metareview: Summary, Strengths And Weaknesses:**

The paper proposes a method to learn implicit manifolds and sample from them using energy-based models and constrained Langevin dynamics. The paper claims that the method can handle submanifolds of any codimension and topologically nontrivial structures. The paper also introduces the concept of manifold arithmetic, which allows performing operations on different manifolds.

The reviewers appreciate the interesting problem of manifold learning and density estimation, and the attempt to combine different techniques from implicit models, energy-based models, and constrained MCMC. However, the reviewers also raise several concerns and criticisms that prevent the paper from being accepted. The main issues are:

- The paper lacks novelty and technical depth. The paper mainly combines existing methods with some minor modifications and generalizations, without providing any theoretical analysis or ablation study to justify the choices and trade-offs. The paper does not compare or contrast its approach with other related work on implicit models, energy-based models, or manifold learning.
- The paper lacks empirical evaluation and validation. The paper only uses synthetic datasets that are either low-dimensional or have known manifold structures. The paper does not demonstrate the advantages of the proposed method over existing methods on real-world datasets or applications. The paper does not provide any quantitative metrics or baselines to measure the performance or quality of the learned manifolds and densities. The paper does not show how the proposed manifold arithmetic can be useful or meaningful for any practical task.
- The paper lacks clarity and motivation. The paper does not clearly explain the background and intuition of the proposed method, especially the energy-based modeling and the constrained Langevin dynamics. The paper does not motivate the problem of manifold learning and density estimation from a practical perspective, or discuss the limitations and challenges of existing methods. The paper does not clearly define the notion of manifold arithmetic or provide any theoretical justification or interpretation for it.

Based on these issues, the reviewers recommend to reject the paper. The paper needs to address the following points to improve its quality and significance:

- Provide more technical details and analysis of the proposed method, and compare it with other related work on implicit models, energy-based models, or manifold learning. Explain the benefits and drawbacks of the proposed method, and provide some theoretical guarantees or insights.
- Conduct more extensive and rigorous experiments on real-world datasets and applications, and provide quantitative metrics and baselines to evaluate the learned manifolds and densities. Demonstrate the superiority of the proposed method over existing methods, and show how the proposed manifold arithmetic can be applied or extended to solve real problems.
- Improve the clarity and motivation of the paper, and provide more background and intuition of the proposed method, especially the energy-based modeling and the constrained Langevin dynamics. Motivate the problem of manifold learning and density estimation from a practical perspective, and discuss the limitations and challenges of existing methods. Define the notion of manifold arithmetic clearly, and provide some theoretical justification or interpretation for it.

---

> ### Author Response · Authors · 2023-02-10
> **Response to Metareview**
>
> Thank you to the reviewers and area chair for their time and work reviewing this paper. We respect the AC’s final decision, but feel the need to point out for future readers that there are some factual inaccuracies in the metareview:
> * “The paper only uses synthetic datasets.” This is not true. We showed how our model performs on real-world geospatial and amino acid data, which are genuine applications.
> * “The paper does not provide any quantitative metrics.” This was not a concern of the reviewers, so we did not raise this in the responses, but there are indeed quantitative metrics in Table 1 of the appendix, with the appropriate caveats about the difficulty of measuring manifold quality.
>
> We also feel it appropriate to cast doubt on one strong subjective statement in the metareview:
> * “The paper lacks novelty and technical depth.” In our paper, we presented an entirely new way to model manifolds with generative models. We built the technical tools to do this from previously unrelated parts of the ML literature. The novelty criticism seems unwarranted to us.